# Microstructures and Mechanical Properties of H13 Tool Steel Fabricated by Selective Laser Melting

**DOI:** 10.3390/ma15072686

**Published:** 2022-04-06

**Authors:** Fei Lei, Tao Wen, Feipeng Yang, Jianying Wang, Junwei Fu, Hailin Yang, Jiong Wang, Jianming Ruan, Shouxun Ji

**Affiliations:** 1State Key Laboratory of Powder Metallurgy, Central South University, Changsha 410083, China; 1533170003@csu.edu.cn (F.L.); 213301068@csu.edu.cn (T.W.); 203301064@csu.edu.cn (F.Y.); jianying.wang@csu.edu.cn (J.W.); jianming@csu.edu.cn (J.R.); 2Key Laboratory of Marine Environmental Corrosion and Bio-Fouling, Institute of Oceanology, Chinese Academy of Sciences, Qingdao 266071, China; hitfujw@163.com; 3Brunel Centre for Advanced Solidification Technology (BCAST), Brunel University London, Uxbridge UB8 3PH, Middlesex, UK; shouxun.ji@brunel.ac.uk

**Keywords:** H13 tool steel, selective laser melting, microstructure, mechanical properties

## Abstract

H13 stool steel processed by selective laser melting (SLM) suffered from severe brittleness and scatter distribution of mechanical properties. We optimized the mechanical response of as-SLMed H13 by tailoring the optimisation of process parameters and established the correlation between microstructure and mechanical properties in this work. Microstructures were examined using XRD, SEM, EBSD and TEM. The results showed that the microstructures were predominantly featured by cellular structures and columnar grains, which consisted of lath martensite and retained austenite with numerous nanoscale carbides being distributed at and within sub-grain boundaries. The average size of cellular structure was ~500 nm and Cr and Mo element were enriched toward the cell wall of each cellular structure. The as-SLMed H13 offered the yield strength (YS) of 1468 MPa, the ultimate tensile strength (UTS) of 1837 MPa and the fracture strain of 8.48%. The excellent strength-ductility synergy can be attributed to the refined hierarchical microstructures with fine grains, the unique cellular structures and the presence of dislocations. In addition, the enrichment of solute elements along cellular walls and carbides at sub-grain boundaries improve the grain boundary strengthening.

## 1. Introduction

Selective laser melting (SLM) is an additive manufacturing (AM) process to make components from melting alloy powder at specific locations layer by layer, which has received significant attention in recent years because of enabling the customized metallic components with complex geometries [1,2,3]. Compared with traditional casting methods, the highly localized melting and ultrafast cooling rate can generate unique non-equilibrium microstructures and superior mechanical properties [4,5].

The H13 tool steel has been widely used in industry for making moulds/dies for elevated applications because of the advantages in high strength, excellent ductility, good wear resistance and machinability [6]. Generally, the moulds/dies feature uniqueness and a low number of production. Therefore, it is very suitable for additive manufacturing. The microstructure and mechanical properties have been studied systematically for the SLMed H13. But the mechanical properties of as-SLMed H13 are often very scattered and unsatisfactory in applications. The variation of YS are normally from 830 to 1342 MPa and that of UTS are from 900 to 1712 MPa in the as-SLMed H13 [6,7,8,9,10,11]. More importantly, the elongation of as-SLMed H13 is normally lower than that obtained by conventional methods [12,13,14]. It was addressed that the high fluctuation of mechanical properties are mainly a response for the different contents of residual austenite [15,16], high residual stresses [17,18,19] and defects [20,21].

To address the key issues of severe brittleness and scatter distribution of mechanical properties of the H13 produced by SLM, various methods have been utilized, including the optimization of process parameters, substrate preheating and heat treatment of as-SLMed H13. The optimization of process parameters is always the key to obtain high density in as-SLMed alloys. Recent works have obtained the densities of ≥99% by modifying scan speed, laser power, powder federate and hatch spacing [22,23,24,25]. The substrate preheating has also been found as an effective technique in eliminating cracks, residual thermal stresses and can prevent the delamination in the as-SLMed H13 steel resulting from the reduction of cooling rates [26,27]. Mertens et al. [26] reported that the as-SLMed H13 showed a homogeneous morphology and superior mechanical properties when preheating the substrate at 400 °C. The post-heat treatment of as-SLMed H13 was also found as an effective method to remove the residual stresses, tune the microstructure and improve the mechanical properties. It has been reported that the as-SLMed tool steel can achieve a significant secondary hardening when tempered at 300–600 °C [28,29,30]. It was found that the as-SLMed H13 could achieve the yield strength of 1483 ± 48 MPa, the high ultimate strength of 1938 ± 62 MPa and the fracture strain of 5.8% after being tempered at 600 ºC [5]. However, heat treatment further increases the cost and potentially destroys the fine microstructure produced by the as-SLMed process, especially the cellular structures. Zhong et al. [31] reported that the microstructures with intergranular cellular segregation network could increase the yield strength without scarifying the ductility. Zhu et al. [32] and Wang et al. [33] demonstrated that the cellular structure significantly increased the strength via dislocation hardening. The strategies to achieve high strength and ductility simultaneously in as-SLMed H13 steel are becoming attractive and critical for industrial applications. However, the strengthening mechanisms have not been well understood for cellular structures, dislocations, microstructural refinement and precipitates in the as-SLMed H13 [34,35,36,37,38]. More technical evidences are needed to confirm the achievement in high strength and ductility for the as-SLMed H13, as well as the detailed strengthening mechanisms.

In this work, we aimed to study the microstructure and mechanical properties of as-SLMed H13. The process optimization, the characterization of powders, the microstructure and mechanical properties in the as-SLMed H13 were systemically studied. The discussion focuses on the correlation between the cellular structures, the mechanical response and the dominant mechanisms responsible for strengthening the as-SLMed H13.

## 2. Experimental

### 2.1. Powder Preparation

H13 powders **were** provided by Hunan Hualiu New Materials Co., Ltd. (Hunan, China). Inductively coupled plasma atomic emission spectrometry (ICAP 7000 Series, Waltham, MA, USA) was used to determine the chemical composition of the H13 powders in Table 1. The sizes of H13 powders were tested via laser particle size analyser (Matersizer, Malvern, UK). The size distribution ranged from 15 to 53 μm, with *D_10_* = 16.8 μm, *D_90_* = 47.3 μm and *D_50_* = 28.4 μm. The powder micrograph and particle size distribution are displayed in Figure 1.

### 2.2. SLM Manufacturing Process

The SLM process was conducted using an FS271M selective laser melting system (Farsoon, Inc, Hunan, China). The samples were processed layer-by-layer on an H13 steel plate with 67° rotating scanning (Refer to [39,40]). The orientation changes for each layer and the angle difference between adjacent layers is 67º, as depicted in Figure 2a. The cubic samples (10 × 10 × 10 mm^3^) and cuboid samples (80 × 10 × 10 mm^3^) were fabricated based on the following processing parameters: hatch spacing of 0.1 mm, layer thickness of 0.03 mm, laser power of 170, 200, 230 and 260 W, scan speed of 600, 800 and 1400 mm/s. Figure 2b shows the samples of cuboid and cubes for microstructure and mechanical property testing. Additionally, the dog-bone-shaped sample for the test was with a spacing length of 30 mm and a cross-section of 4 × 2 mm^2^ is shown in Figure 2c. Correspondingly, a representative schematic diagram showing the experimental procedures of H13 steel is illustrated, as shown in Figure 3.

### 2.3. Microstructural Characterization

The densities of as-SLMed H13 samples were tested via Archimedes method [5]. All the featured data are based on the mean value of at least 5 measurements. The specimen density was calculated by the following equation:(1)$$\rho =\frac{\rho_{0}m}{{m\;-\;m}_{0}}$$
where *ρ* is specimen density (g/cm^3^), *ρ_0_* is distilled water density (g/cm^3^), *m* is specimen weight in air and *m_0_* is specimen weight in distilled water. The relative density was obtained by comparing the specimen density with the theoretical density.

The volumetric energy density (*VED*) was computed via the Equation (2):(2)$$VED=\frac{P}{v\;\times \;h\;\times \;l}$$
where *P* is laser power (W), *v* is scanning speed (mm/s), h is hatch spacing (mm), and *l* is layer thickness (mm). Figure 4 shows the relative density of as-SLMed H13 specimens with *VED*.

The phase constituent crystal structure of the as-SLMed H13 was examined in terms of X-ray diffraction with Cu Kα radiation (XRD, SmartLab 3Kw, Rigaku, Tokyo, Japan). Specimens for microstructural observations were etched with 4 vol% nitric acid solution. Microstructural features were characterized using a scanning electron microscopy (SEM, Quanta 250 FEG, FEI, Brno, Czech Republic). The grain size and grain orientation were detected by using electron backscattered diffraction (EBSD, Helios NanoLab G3 UC, FEI, Hillsboro, OR, USA) equipped with the TSL OIM data analysis. Furthermore, the transmission electron microscope (TEM; Tecnai G2F20, FEI, Hillsboro, OR, USA) was used for detailed microstructure examination. TEM specimens were fabricated using the precision ion polishing system (PIPS, Gatan691, Gatan, Pleasanton, CA, USA) at a voltage of 5 kV and an incident angle of 3–8°.

### 2.4. Evaluation of Mechanical Properties

Micro-hardness was measured using a micro-Vickers hardness instrument (HMV-2T, Shimadzu, Kyoto, Japan) with 300 g load for 15 s, and the average value was taken from at least 8 points of each sample. The dog-bone-shaped tensile specimens were cut by electrical discharge machining (EDM) from the as-SLMed specimens. Uniaxial tensile tests were evaluated via material testing system (Alliance RT30, MTS, Shanghai, China) with an engineering strain rate of 1 × 10^−3^ s^−1^ at room temperature. The tensile data were the mean of at least 5 measurements.

## 3. Results

### 3.1. Mechanical Properties

Figure 5 shows micro-hardness of the as-SLMed H13 samples. Obviously, the micro-hardness increases and then decreases with increasing VED. In addition, the micro-harnesses were lower in the building direction than those in the horizontal direction. The peak micro-hardness was obtained at the VED of 95.8 J/mm^3^ and the corresponding micro-hardness was 537.5 Hv in the building and 560.9 Hv in the horizontal direction. According to the relative density and micro-hardness, the optimal process parameters were: *P* = 230 W, *Vs* = 800 mm/s and *VED* = 95.8 J/mm^3^.

Figure 6 shows the tensile stress-strain curves of the as-SLMed H13 samples. The mechanical properties are listed in Table 2. It is seen that an excellent combination with the YS of 1468 MPa, the UTS of 1837 MPa and the elongation of 8.5% were achieved at the VED of 95.8 J/mm^3^. Figure 6b shows a comparison of tensile properties of H13 steel specimens fabricated by different techniques, including conventional casting methods [8,9,41], SLM [6,7,10,13], SLM + heat treatment [5,8]. Obviously, the method used in this work with optimised parameters could produce the optimal combination of strength and ductility for H13 steel.

Figure 7 shows the fractured morphologies of the as-SLMed H13 specimens processed at the VED of 95.8 J/mm^3^. Little defects, such as unmelted powder and microvoids in the matrix at low magnification, were observed. A large number of dimples were observed in Figure 7b, which confirmed the existence of brittle and ductile fracture mixture.

### 3.2. Microstructural Characterization

Figure 8 shows the XRD spectra obtained from H13 powder and as-SLMed H13 specimens at the VED of 95.8 J/mm^3^. The XRD patterns only presented the peaks of bcc-structured martensite phase (α-Fe) in the H13 steel powder. However, the peaks of bcc-structured martensite (α-Fe) and fcc-structured retained austenite phase (γ-Fe) were detected in the as-SLMed H13 specimens. The presence of residual austenite can be attributed to the fact that SLM process offers a rapid solidification, resulting in the incomplete transformation from austenite to martensite [42,43].

Figure 9 shows the optical micrographs (OM) along horizontal direction and building direction with isometric view. In the horizontal direction, continuous laser tracks were formed with 67° rotating scanning (Figure 9a,b). The melt poor boundaries were clearly observed along the build direction (Figure 9c,d). The defects were likely formed by the entrapment of inert gas or the evaporation of alloy elements. These micropores decreased the strength and elongation because of the easy crack initiation in the pore edge under loading [21]. To further characterize the features of as-SLMed H13, Figure 10 shows the detailed microstructure. The typical microstructure consisted of melt pool (MP) coarse, MP fine and heat affected zone (HAZ). The HAZ was considered as a transition zone between MP coarse and MP fine zones. The columnar grain and equiaxed grain are shown in Figure 10b,c. The equiaxed grain area was observed in the cellular structure. Lath martensite are shown in Figure 10d. The size of cellular structure was about 500 nm.

Figure 11 shows the inverse pole figure (IPF) maps of the as-SLMed H13 specimens. In the horizontal direction (Figure 11a) and building direction (Figure 11b), the IPF maps showed that the microstructure of the as-SLMed H13 specimens was dominated by fine grains with random distribution. In Figure 11c, the martensite content and residual austenite content were 94% and 6% along horizontal direction, respectively. However, it is noted that the residual austenite content of specimens along the building direction (8.7%) was higher than that specimens the along the horizontal direction (6%). The difference in content of residual austenite could be attributed to the different cooling rates during the SLM process [44]. In Figure 11e,g, it is also seen that the mean grain size of martensite and retained austenite were 1.67 μm and 1.15 μm along the horizontal direction, respectively. Meanwhile, the mean grain size of martensite and retained austenite in Figure 10f,h were 1.97 μm and 0.86 μm along the building direction, respectively. The variation is induced by the different cooling rates during processing and different phase contents formed in the alloy. Figure 12 presents the Kernel Average Misorientation (KAM) maps of the as-SLMed H13 specimens and misorientation angle with low-angle grain boundaries (LAGBs) and high-angle grain boundaries (HAGBs). The average KAM value of the building direction and the horizontal direction of the as-SLMed H13 specimens were 0.863 and 0.971, respectively. The variations in average KAM qualitatively reflected the degree of plastic deformation or defect density. According to Equation (1), the KAM was applied to calculate geometrically necessary dislocation (GND):*ρ_GND_* = 2*ϑ*/(*μb*)(3)
where *ϑ* is the average value of KAM, *μ* is the step size of EBSD measurement (0.2 μm), and *b* is the Burger vector (0.25 nm [5]). Correspondingly, *ρ_GND_* of as-SLMed H13 steel along the horizontal direction and building direction were estimated to be ~3.45 × 10^16^ m^−2^ and ~3.88 × 10^16^ m^−2^, respectively. The high density of GNDs would build a solid foundation for high strength. The distributions of misorientation angles map in Figure 12c indicates that the factions of HAGB (≥15°) and LAGB (<15°) were 74.5% and 25.5% along the building direction, respectively.

## 4. Discussions

### 4.1. Relationship in between Microstructure and Mechanical Properties

In general, the SLMed H13 is typically featured by martensite as matrix with a small amount of retained austenite, similar to the conventional casting. Correspondingly, the bcc-structured martensite and the fcc-structured residual austenite in the [$\bar{1}11$] direction are presented in Figure 13a,b. Due to the different cooling rates in different regions of the specimens and the cyclic thermal effect of the continuous layers, high volume fraction of subcooled austenite distributed in the melt pools can be rapidly cooled below the martensite transformation temperature (M_f_) before martensite transformation, and therefore the transformation of some martensite to retained austenite is understandable. Although the generation of retained austenite reduce the strength, the retained austenite induces martensitic transformation during the stretching process, which absorbs stress and reduces the sprouting of cracks. As a result, the tensile strength and toughness are improved [45,46].

Compared to H13 alloys fabricated by other methods such as casting, the corresponding strengthening contributions of H13 alloys mainly depend on microstructure refinement, cellular structure, etc. In Figure 11e, the as-SLMed H13 specimens mainly contain fine grains with a mean grain size of 1.67 μm. Previous studies have shown that the average grain sizes of as-cast H13 steel, as-spray formed H13 steel and as-SLMed H13 steel are in the range of 100~150 μm, 10~20 μm and 2~3 μm [5], respectively. Thus, the contribution of grain boundary strengthening is higher than that of the H13 alloys fabricated by casting and spray forming. Most importantly, the formation of the unique cellular structures in the SLMed H13 alloys possesses outstanding strength. In Figure 10d and Figure 13c, the size of these cellular structures in H13 steel is less than 500 nm. The size and volume fraction of cellular structures varied from sample to sample, in consistent with previous studies [47]. Meanwhile, the solidification conditions are closely related to the formation of cellular structures [48]. Based on the constitutional supercooling theory, the morphology features depends on the thermal gradients (*G*) and the grain growth rates (*R*). Specifically, with a decreasing of the ratio of *G*/*R*, the morphology transforms from isometric to cellular, columnar dendritic or equiaxed dendritic. Therefore, the solidification condition prefers to promote the formation of cellular structures during SLM processing. The ability of cellular structures to improve the strength of as-SLMed alloys has been demonstrated. Wang et al. [49] showed that the cellular structure was the main contributor to the increment of YS of as-SLMed 316L steel. Particularly, the average diameter of cellular structure has been used to measure the strength [34]. To further evaluate the element distribution of the cellular structure, the detail high-angle annular dark field scanning TEM (HAADF-STEM) images are shown in Figure 13d,g. However, due to the strong magnetic properties of the steel sample, the mapping results are not sufficiently clear. It is easy to distinguish that the existence of compositional enrichment of Cr and Mo along the cellular structure, which is consistent with previous investigations [50,51]. Furthermore, in combination with Figure 13e, it can be seen that the part of carbides pinned on the sub-grain boundary, hindering the grain boundary and dislocation movement, and the other part of carbides are uniformly distributed in the sub-grain boundaries as second phase particles. Finally, dislocations can be retained at grain boundaries, as shown in Figure 13f.

### 4.2. The Strengthening Mechanisms

The results showed that the as-SLMed H13 steel processed at the VED of 95.8 J/mm^3^ exhibits the YS of 1468 MPa, UTS of 1837 MPa and fracture strain of 8.48%. Previous studies have shown that the cellular structure strengthening (σ_c_), fine grain strengthening (σ_g_) and dislocation strengthening (σ_dis_) are the main contributing factors of the yield strength of the as-SLMed steel [5,49,52]. In this study, the cellular structure, the fine grain size (~1.67 μm) and the dislocations induced by the thermal contraction stress during the rapid solidification of SLM process are also the main contributors of high yield strength. Additionally, the precipitation strengthening induced by carbides could be negligible due to the very low volume fraction. Thus, the contribution of the main strengthening contributions of as-SLMed H13 can be calculated as
σ_y_ =σ_0_ + σ_c_ + σ_g_ + σ_dis_(4)
where σ_0_ is a constant (70 MPa for H13 alloy [53]). The cellular structure followed a Hall-Petch type of strengthening behaviour, where the yield strength contributed by the cellular structure (σ_c_) with the average cell diameter (L_c_).
(5)$$\sigma c\;=183.31\;+\;253.66/\sqrt{\mathrm{Lc}}\;$$

The corresponding Lc was estimated as ~500 nm, as shown in Figure 10d and Figure 13c. The strength contribution of the cellular structure to yield strength is estimated as 542 MPa. The strengthening contribution of fine grain strengthening (σg) can be estimated by the Hall-Petch equation:σ_g_ = *Kd*^−0.5^(6)
where *K* is the Hall-Petch constant, and *d* is the average grain size. The *k* (572 MPa μm^1/2^) is taken from literature [37,54]. From Figure 10e, the average grain size of as-SLMed H13 is 1.67 μm. Thus, the contribution of grain refinement to yield strength is calculated as 442.6 MPa. As shown in Figure 13e,f, the interaction between dislocation and wall of cellular structure impedes grain boundary motion and thus enhances strengthening. The contribution of σ_dis_ can be evaluated via Taylor’s hardening law [55]:σ_dis_ = *M∙α∙G∙b∙ρ*^0.5^(7)
where *M* is the Taylor factor (3 for BCC crystal structure of iron [55]), *α* is a material constant (0.33 [55]), *G* is the shear modulus (80 GPa [55]), *b* is the Burgers vector (0.25 nm [55]), and *ρ* is the dislocation density (taken to the horizontal direction). The strength contribution of dislocation density to yield strength is estimated as 367.8 MPa. The estimated yield strength is ~1422.4 MPa (σ_0_ ≈ 70 MPa; σ_c_ ≈ 542 MPa; σ_g_ ≈ 442.6 MPa; σ_dis_ ≈ 367.8 MPa), it is consistent well with the experimentally data of 1468 MPa. Furthermore, the high yield strength of H13 steel should also originate from other factors. For example, a small number of carbides distributed in and out of sub-grain boundaries were found for the as-SLMed H13 steel. Verifiably, the yield strength of the as-SLMed H13 specimens is mainly contributed by the strengthening from cellular structure and grain-refinement, followed by dislocation strengthening.

## 5. Conclusions

In this study, microstructure and mechanical properties of the as-SLMed H13 have been investigated. Our major conclusions are as follows:

(1) H13 steel could be fabricated successfully by SLM in this study. The maximum relative density reached 99.6% under the optimized process parameters, including *P* = 230 W, *Vs* = 800 mm/s and *VED* = 95.8 J/mm^3^.

(2) The as-SLMed microstructure consists of lath martensite and 6~8.7% retained austenite. The as-SLMed structure with a typical cellular size of ~500 nm is predominantly cellular morphology and Cr and Mo element were enriched toward the cell wall of each cellular structure.

(3) The as-SLMed H13 can offer a remarkable synergistic improvement in strengthen and ductility, in which the yield strength is 1468 MPa, the UTS is 1837 MPa and the fracture strain is 8.48%. Numerous dimples are found in the fractured surface, indicating the existence of ductile fractures.

(4) The excellent strength-ductility synergy can be attributed to the refined hierarchical microstructures with fine grains at an average size of 1.67 μm, cellular structures and dislocations. The calculated contribution to the yield is 542 MPa for the cellular structure strengthening, 442.6 MPa for the grain-refinement strengthening, and 367.8 MPa of dislocation strengthening.

## Figures and Tables

**Figure 1 materials-15-02686-f001:**
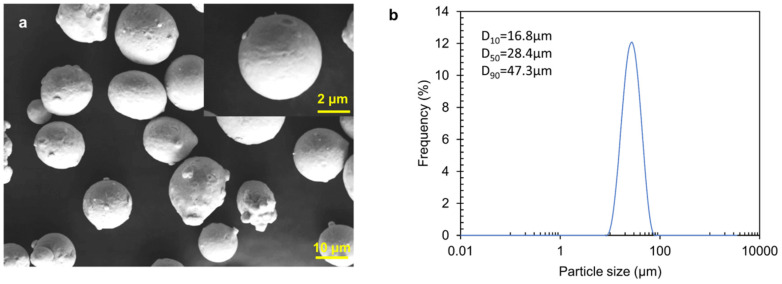
(**a**) SEM micrograph showing the micrograph of H13 powders; (**b**) distribution of particle sizes.

**Figure 2 materials-15-02686-f002:**
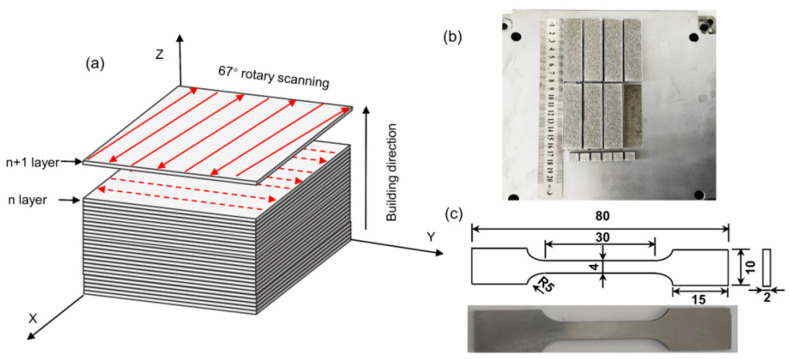
(**a**) Schematic diagram showing laser scanning approach during additive manufacturing; (**b**) Produced samples of cuboid and cubes; (**c**) Sample size for tensile test.

**Figure 3 materials-15-02686-f003:**
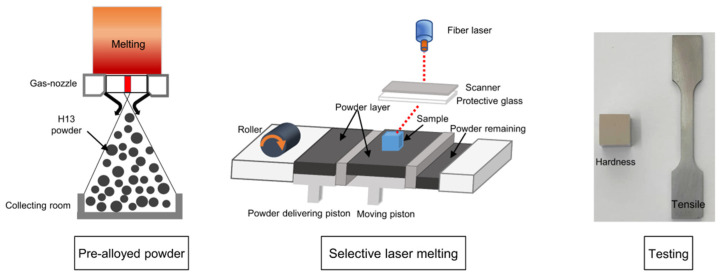
Schematic diagram showing the experimental procedures of H13 steel, including powder preparation, selective laser melting and the geometry of testing samples.

**Figure 4 materials-15-02686-f004:**
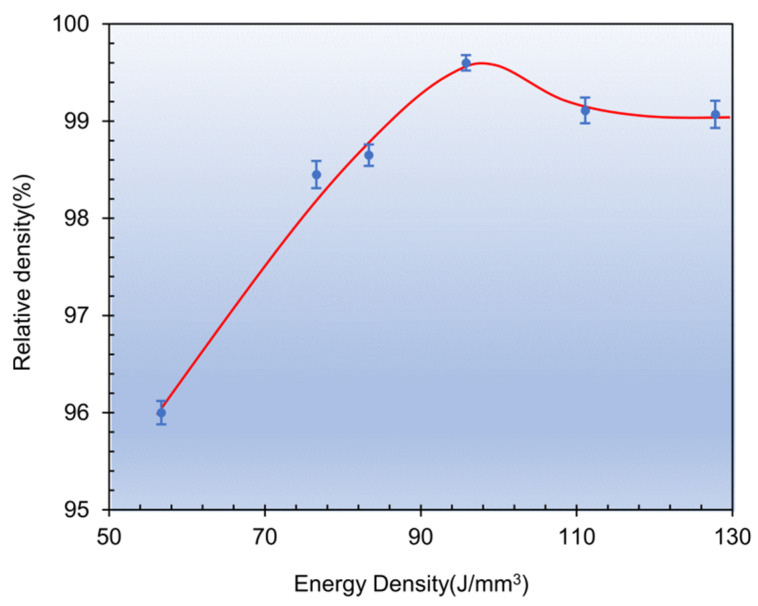
The relative density of as-SLMed H13 samples with different VEDs.

**Figure 5 materials-15-02686-f005:**
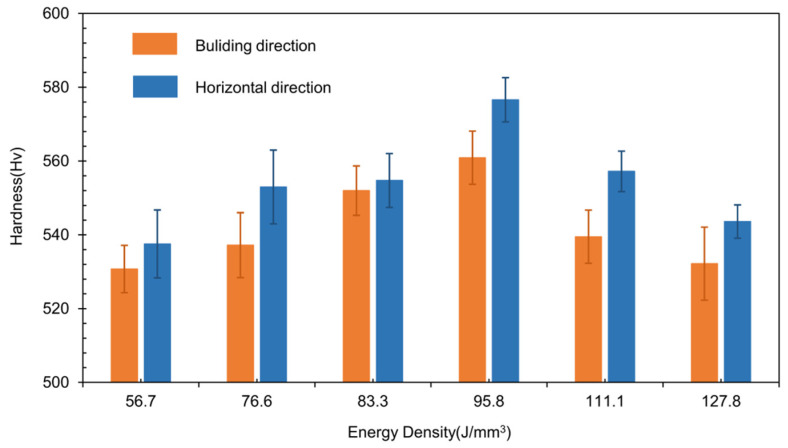
Micro-hardness of the H13 specimens with different VED.

**Figure 6 materials-15-02686-f006:**
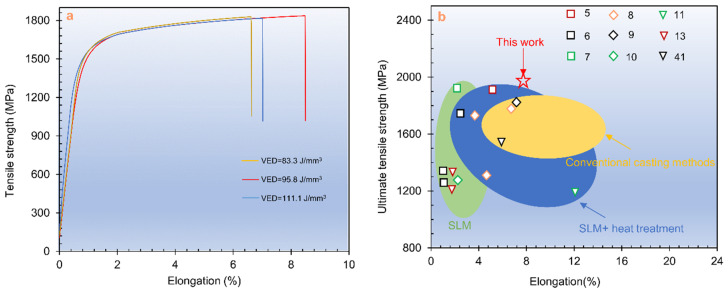
(**a**) Tensile stress–strain curves of SLMed H13; (**b**) Comparison of tensile strength of the H13 specimens manufactured by different techniques, including conventional casting methods, SLM, SLM + heat treatment.

**Figure 7 materials-15-02686-f007:**
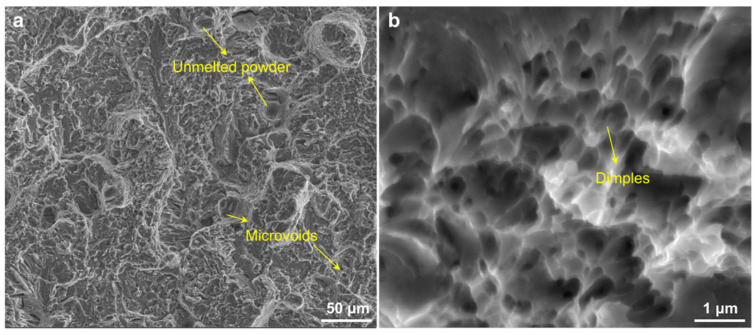
(**a**) SEM micrographs showing the unmelted and microvoids on fracture of tensile specimens possessed at the VED of 95.8 J/mm^3^; (**b**) Detailed fracture morphology showing fine dimples.

**Figure 8 materials-15-02686-f008:**
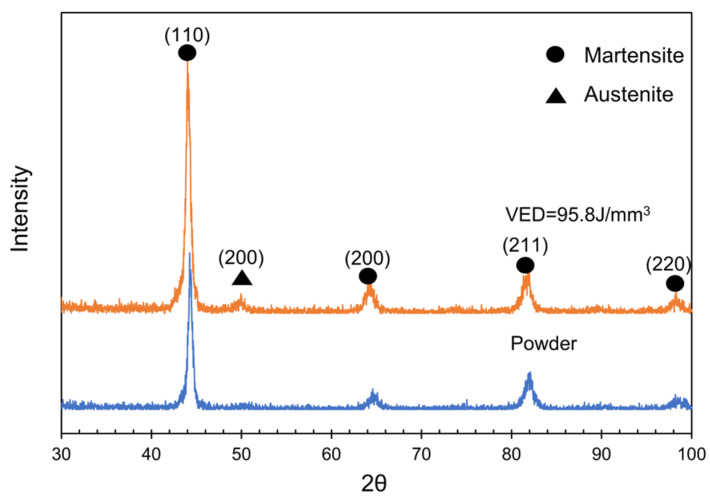
XRD spectra of H13 powder and as-SLMed H13 specimens at the VED of 95.8 J/mm^3^.

**Figure 9 materials-15-02686-f009:**
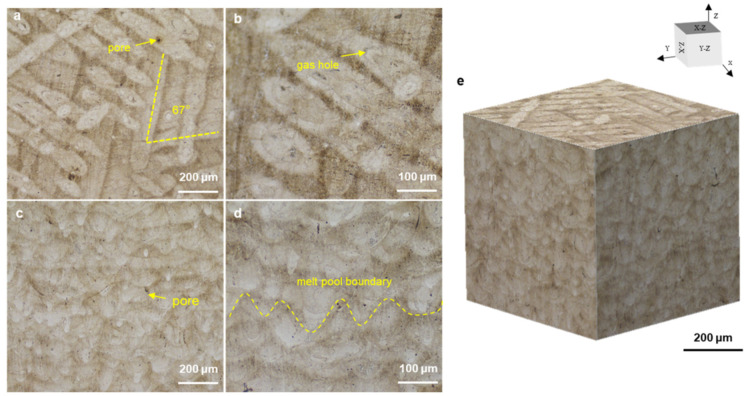
Optical micrographs showing the microstructure along (**a**,**b**) horizontal direction and (**c**,**d**) building direction; and (**e**) isometric view.

**Figure 10 materials-15-02686-f010:**
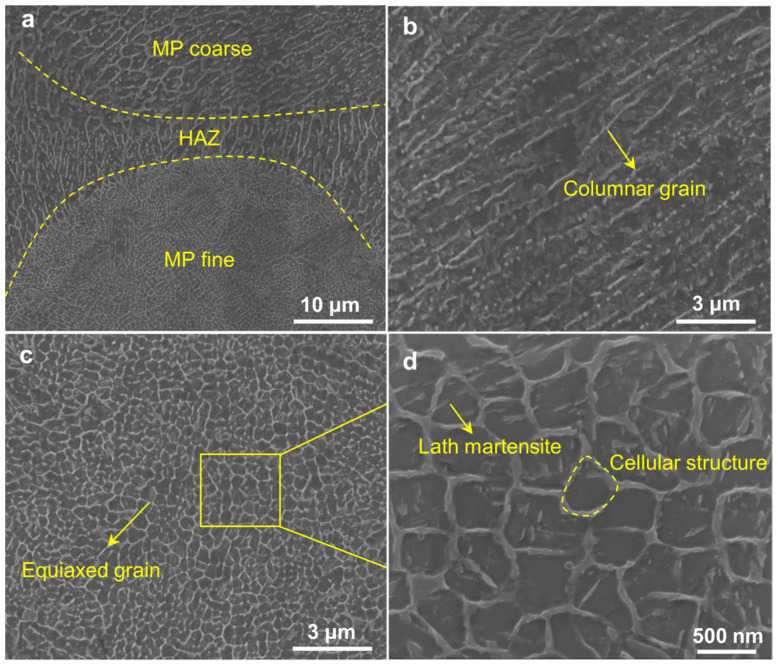
SEM images displaying the microstructure features of as-SLMed H13 specimens, (**a**) molten pool structure; (**b**) columnar grains; (**c**) equiaxed grains; (**d**) cellular structures and lath martensite.

**Figure 11 materials-15-02686-f011:**
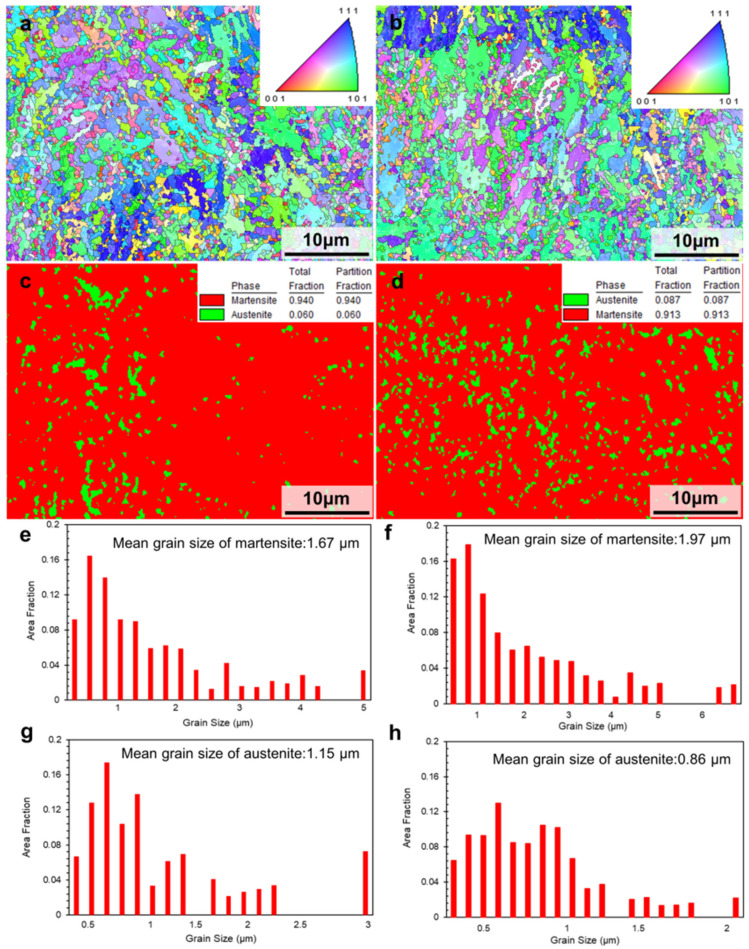
Inverse-pole figure (IPF) maps of the as-SLMed H13 specimens showing the grain orientations, phase maps and grain size distribution maps: (**a**,**c**,**e**,**g**) horizontal direction; (**b**,**d**,**f**,**h**) building direction.

**Figure 12 materials-15-02686-f012:**
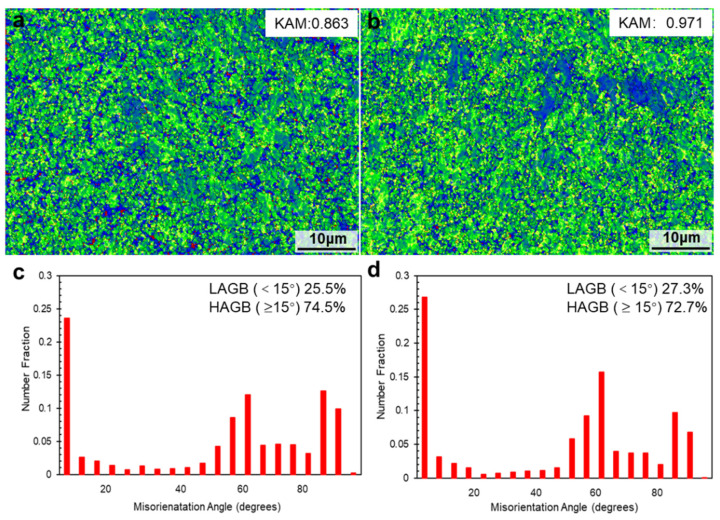
Kernel average misorientation (KAM) maps of the as-SLMed H13 specimens and misorientation angle with low-angle grain boundaries (LAGBs) and high-angle grain boundaries (HAGBs): (**a**,**c**) horizontal direction; (**b**,**d**) building direction.

**Figure 13 materials-15-02686-f013:**
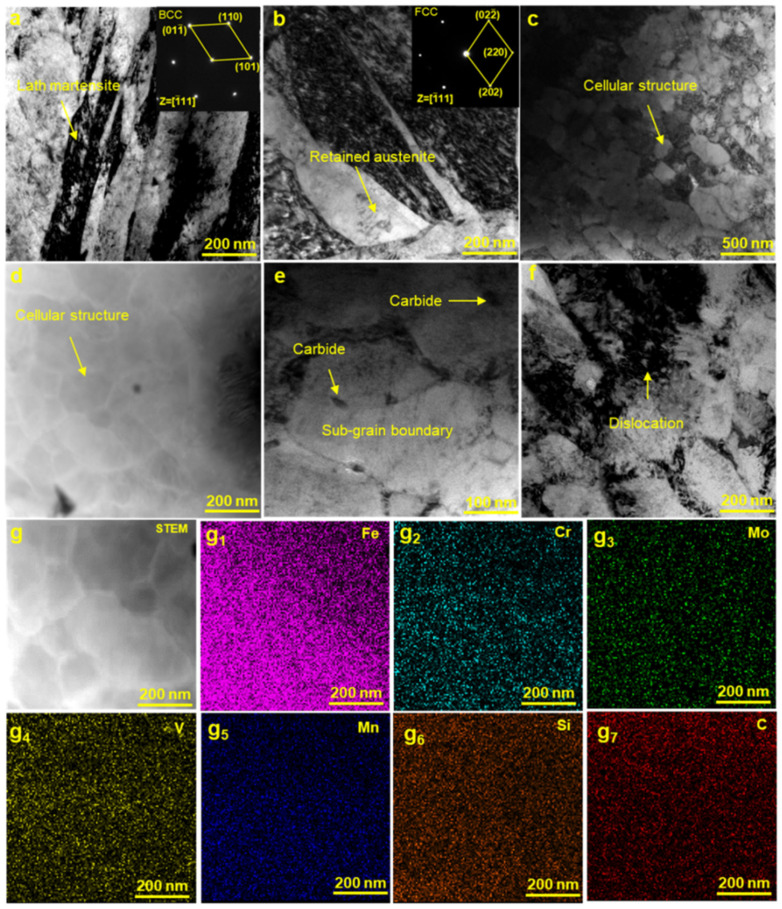
TEM micrographs of the as-SLMed H13 steel specimens, showing (**a**) bcc-structured lath martensite; (**b**) the formation of retained austenite with high cooling rate; (**c**) cellular structure; (**d**) STEM image of the cellular structure shown in (**c**); (**e**) carbides at and within sub-grain boundaries; (**f**) the density of dislocations; (**g**, **g**_1_–**g**_7_) the corresponding EDS maps of cellular structure, including elements of (**g**_1_) Fe, (**g**_2_) Cr, (**g**_3_) Mo, (**g**_4_) V, (**g**_5_) Mn, (**g**_6_) Si and (**g**_7_) C.

**Table 1 materials-15-02686-t001:** Chemical Compositions of the H13 powders (wt.%).

Element	Cr	Mo	V	Mn	Si	C	Fe
wt.%	5.12	1.26	1.03	0.39	0.98	0.42	Bal.

**Table 2 materials-15-02686-t002:** Mechanical properties of the SLMed H13 samples under different conditions.

VED	YS (MPa)	UTS (MPa)	Elongation (%)
83.3 J/mm^3^	1456 ± 32	1828 ± 48	6.6 ± 0.3
95.8 J/mm^3^	1468 ± 27	1837 ± 23	8.5 ± 0.6
111.1 J/mm^3^	1408 ± 29	1817 ± 31	7.0 ± 0.4

## Data Availability

The data presented in this study are available on request from the corresponding author.

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
