# Peer review of "Microstructures and Mechanical Properties of H13 Tool Steel Fabricated by Selective Laser Melting"

_materials, 2022, doi:10.3390/ma15072686_

Round 1

Reviewer 1 Report

 Reviewer Comments

In this study, an experimental investigation is performed on the microstructure and mechanical properties of H13 alloy processed by selective laser melting. The topic is interesting and addresses today's industrial problems, however, there are several technical questions and comments. In this regard, considering that the topic of the work is interesting, I would like to suggest Major Revision, based on the following comments. The English language of the manuscript text requires serious double-checking.

Please respond to each comment in the file of “author’s response to comments” and include the page number (in the manuscript) for each response, while highlighting all changes in the new version of the manuscript.

My comments:

  1. In the abstract, please write the full name for “SLMed”
  2. The abstract should be upgraded in a structured form to have a short sentence of 1) Introduction, 2) Objective, 3) Materials and Method, 4) Result and Discussion, and 5) Conclusion.
  3. The first paragraph of section 2 (before subsection 2.1) describes figure 1. It seems you are trying to explain the experimental procedure, but it is not proper to start a section like this. You must re-sentence and provide sufficient about the experimental process that explains the detail while pointing to figure 1.
  4. Figure 1 should be updated with a real microscopic image of the sample on the right side.
  5. Table 1 requires references
  6. Page 4; you need to explain what is “FS271M SLM system”
  7. Figure 3(a); is the rotary direction of the second layer 0-degree? Why is that? If the orientation changes for each layer, please explain in the manuscript.
  8. Figure 3(b); you must show a microscopic image of the sample that show the layers
  9. Page 4, 1st paragraph: why 67° rotating scanning? Why not other angles? If you need to cite some references please do so
  10. Page 5: you need to explain the “Archimedes method” and add a reference
  11. Subsection 2.4, 1st line: The micro-hardness test WAS PERFORMED using….
  12. Subsection 2.4: did you cut the dog-bone-shaped samples from samples of figure 3(b). you must show a dog-bone sample beside figure 3(b)
  13. I can't find the full name of VED
  14. Figure 5; how did you measure the energy density? please explain in the text
  15. Page 6, last paragraph: please name the “different techniques”
  16. Figure 6a; is each curve the average result of many tests? Please explain in the manuscript
  17. Figure 7(a) is not clear, can you show several unmelted powders?
  18. The caption of figure 9 doesn’t need (OM)
  19. Figure 10 is very valuable
  20. Figure 13f is not clear to show a large number of dislocation
  21. Page 12: the sentence “It is also found the more refined grains can be obtained in this work” is vague.
  22. Page 13, 2nd line: It is easily EASY to distinguish
  23. There are too many grammatical mistakes and the reviewer could not point out all of them
  24. How are you confident to propose equation 2?
  25. Page 14: what is the physical/mechanical means of “dislocation strengthening”? do you mean resistance to the dislocation process?!
  26. Equation 5 and Taylor’s hardening law need a reference
  27. Section Conclusion must first explain the manuscript topic and what you have done, then point out the finding and conclude

Author Response

Please find it in the attachement.

Reviewer 2 Report

1- Fig 2(b) is about the distribution of particle sizes, it may be explained in the draft more clearly.

2- Please write the expanded form with its abbreviation the first time, e.g., VED: Volumetric Energy Density .

3- There is not explanation about the EDX results at Fig.10, Fig 13g. what was the purpose of this characterization? please clarify it clearly?

4-  Conclusion does not include all the highlighted points in the draft. It should be revised.

Author Response

Please find it in the attachement.

Round 2

Reviewer 1 Report

The manuscript is revised well, I have only a few comments”

  1. Regarding my comments " Figure 3(a); is the rotary direction of the second layer 0-degree? Why is that? If the orientation changes for each layer, please explain in the manuscript."; please explain why did you select the 67deg angle and the philosophy behind it. 
  2. In response to my question “ Figure 3(b); you must show a microscopic image of the sample that shows the layers.”; you mentioned “you are not able to take a microscopic picture” then how are you sure if the layers are really made as you claim?
  3. Regarding the comment “ Page 5: you need to explain the “Archimedes method” and add a reference.”, I recommend adding your response in the manuscript text.
  4. Your response to the two comments “ How are you confident to propose equation 2?” and “25. Page 14: what is the physical/mechanical means of “dislocation strengthening”? Do you mean resistance to the dislocation process?” should be adequately placed in the manuscript text.
  5. Please carefully consider MDPI English language service to improve the manuscript text grammar and technical explanation of the results. 
